Comprehensive tissue-specific transcriptome profiling of pineapple (Ananas comosus) and building an eFP-browser for further study

Mao Qi 1 2
Chen Chengjie 1
Xie Tao 1
Luan Aiping 3
Liu Chaoyang 1
He Yehua 1 heyehua@scau.edu.cn
1 Key Laboratory of Biology and Germplasm Enhancement of Horticultural Crops in South China, Ministry of Agriculture, College of Horticulture, South China Agricultural University , Guangzhou , China
2 College of Agriculture, Guangdong Ocean University , Zhanjiang , China
3 Tropical Crops Genetic Resources Institute of Chinese Academy of Tropical Agricultural Science , Danzhou , China
VanBuren Robert
Electronic publication date: 2018 Dec 4
Publication date: 2018
Volume: 6
Electronic Location ID: e6028
Received 2018 Jul 30; Accepted 2018 Oct 29
Copyright: © 2018 Mao et al.
Copyright year: 2018
Copyright holder: Mao et al.
License: This is an open access article distributed under the terms of the Creative Commons Attribution License, which permits unrestricted use, distribution, reproduction and adaptation in any medium and for any purpose provided that it is properly attributed. For attribution, the original author(s), title, publication source (PeerJ) and either DOI or URL of the article must be cited.
License URL: https://creativecommons.org/licenses/by/4.0/

Keywords: Pinapple, Fruit, Tissue-specific genes, eFP-browser, Transcriptome

Funding: Technology Commission of Guangdong Province 2013B020304002 National Key R&D Program of China 2018YFD100504 Foundation of Young Creative Talents in Higher Education of Guangdong Province 2017KQNCX020 National Natural Science Foundation of China 31572089 and 31801831 Modern Agricultural Industry Technology System of Guangdong Province 2016LM1128 Central Public-interest Scientific Institution Basal Research Fund for Chinese Academy of Tropical Agricultural Sciences No. 1630032018015 This study was supported by the Technology Commission of Guangdong Province (2013B020304002), the National Key R&D Program of China (2018YFD100504), the Foundation of Young Creative Talents in Higher Education of Guangdong Province (2017KQNCX020), the National Natural Science Foundation of China (31572089 and 31801831), the Modern Agricultural Industry Technology System of Guangdong Province (2016LM1128) and the Central Public-interest Scientific Institution Basal Research Fund for Chinese Academy of Tropical Agricultural Sciences (No. 1630032018015). The funders had no role in study design, data collection and analysis, decision to publish, or preparation of the manuscript.

==============================
Pineapple is one of the most economically important tropical or subtropical fruit trees. However, few studies focus on the development of its unique collective fruit. In this study, we generated a genome-wide developmental transcriptomic profile of 14 different tissues of the collective fruit of the pineapple covering each of the three major fruit developmental stages. In total, 273 tissue-specific and 1,051 constitutively expressed genes were detected. We also performed gene co-expression analysis and 18 gene modules were classified. Among these, we found three interesting gene modules; one was preferentially expressed in bracts and sepals and was likely involved in plant defense; one was highly expressed at the beginning of fruit expansion and faded afterward and was probably involved in endocytosis; Another gene module increased expression level with pineapple fruit development and was involved in terpenoid and polyketide metabolism. In addition, we built a pineapple electronic fluorescent pictograph (eFP) browser to facilitate exploration of gene expression during pineapple fruit development. With this tool, users can visualize expression data in this study in an intuitive way. Together, the transcriptome profile generated in this work and the corresponding eFP browser will facilitate further study of fruit development in pineapple.

Introduction

Pineapple is an economically important fruit herb widely cultivated in tropical and subtropical regions (Sanewski, Bartholomew & Paull, 2018). Pineapples are unique agricultural fruit not only due to their fragrance and taste, but also because they form collective fruits, which differ them from other familiar fruits such as apples, oranges, and mangoes. All fleshy coalesced floral parts of pineapple flowers and its inflorescence can be consumed as either fresh or canned fruit (Sanewski, Bartholomew & Paull, 2018; Botella & Smith, 2008). Thus, pineapple fruit possesses both economic and research value.

Recently, with the popularization of high-throughput sequencing technologies, several genomic studies of pineapple fruit have been performed (Moyle et al., 2005; Ong, Voo & Kumar, 2012; Redwan, Saidin & Vijay Kumar, 2016). Moyle et al. (2005) analyzed expressed sequence tags (EST) clone sequences and found that contigs from libraries of green (unripe) and yellow (ripe) fruit were both dominated by metallothionein. To understand the mechanisms underlying fruit ripening, Ong et al. performed Illumina sequencing on the transcriptome of ripe pineapple fruit flesh. Genes related to flavor, texture, appearance, and fruit sweetness were identified based on digital annotation (Ong, Voo & Kumar, 2012). In addition, Redwan, Saidin & Vijay Kumar (2016) analyzed differential gene expression in RNAseq libraries from ripening pineapple fruit; this analysis revealed that ethylene-related genes may also be involved in regulating non-climacteric ripening in pineapple. While each of these studies examined pineapple fruit, they were all concerned with fruit ripening. While understanding fruit ripening is highly important, so is fruit development, which occurs immediately prior to ripening. How does each pineapple fruit expand? How is the collective fruit formed? How can we breed a cultivar with smaller and shallower “fruit eyes”? Answering these questions requires a better understanding of pineapple fruit development, and especially a better understanding of how different tissues merge to make up collective fruits.

The electronic fluorescent pictograph (eFP) browser is a powerful web-based tool for exploring gene expression (Winter et al., 2007). It displays cartoon images depicting various tissue types in various colors, with each hue indicating the level of expression of a queried gene at a given time point (Hawkins et al., 2017). Since the development of the original eFP browser for Arabidopsis, eFP browsers have also been developed for other plants, including poplar, soybean, potato, maize, rice, and barley (http://www.bar.utoronto.ca/). In addition, eFP browsers have also been developed for several fruit-bearing horticultural crops, including tomato (http://bar.utoronto.ca/efp_tomato/cgi-bin/efpWeb.cgi), grape (http://bar.utoronto.ca/efp_grape/cgi-bin/efpWeb.cgi), and strawberry (Hawkins et al., 2017).

Databases make bioinformation accessible to research community. Moyle et al. (2005) developed “PineappleDB,” providing access to expressed sequences tag isolated from pineapple fruit, root, and nematode infected root gall vascular cylinder tissues. Recently, Xu et al. (2018) have reported of an integrated pineapple genomics database (PGD) storing gene information resources of pineapple. PGD is an excellent database for the research community, which covering 45 RNAseq samples generated from pineapple genome sequencing project that mainly focuses on crassulacean acid metabolism study (Ming et al., 2015). Currently, there is no eFP browser available for pineapple. This may be because most experiments have been performed treating the whole pineapple fruit as a single tissue and thus no eFP browser was needed. However, to investigate the development of pineapple fruit, a scheme whereby different tissues—for example, receptacles, ovary walls, and placentas—can be examined independently is more appropriate. In this study, we obtained samples of 11 tissues of the pineapple fruit at three developmental stages, as well as three vegetative tissues. We then generated RNAseq libraries for all samples, resulting in 36 libraries. After cautious transcriptomic profiling, we set up the first eFP browser for pineapple (available at: http://www.tbtools.xyz/efp/cgi-bin/efpWeb.cgi), aiming to provide an important resource for researchers interested in pineapple breeding and crop improvement.

Methods and Materials

Plant material preparation and RNA sequencing

All plants of pineapple cultivar “Shenwan” used for this study were cultivated in Shenwan Town, Zhongshan City, Guangdong, China. Three distinct plant stages were defined according to fruit development. Stage 1 begins when flower buds begin to bloom and the collective fruit begins to expand in size. Stage 2 begins when flowering ceases, and the collective fruit expands at its fastest rate. Stage 3 corresponds to the time when collective fruit stops expanding and begins to ripen. In total, 11 tissues were manually separated from pineapple. These included bract, core, flower disk, leaf, ovary wall, ovule, petal, placenta, receptacle, sepal, stamen, and style tissues. All samples were collected and stored immediately in liquid nitrogen before RNA extraction. We extracted total RNA from all samples using TRIzol (Invitrogen, Carlsbad, CA, USA) and determined RNA quality of samples using Agilent 2100. RNA samples with RIN numbers lower than eight were re-extracted. All RNA samples were then forwarded to a sequencing provider for RNA sequencing.

Expression profiling

Pre-filtered data received from the sequencing provider was manually checked by FastQC (http://www.bioinformatics.babraham.ac.uk/projects/fastqc/). All sequencing adapters and low quality reads were filtered using Trimmomatic (Bolger, Lohse & Usadel, 2014). Only paired reads were kept for downstream analysis. STAR aligner was employed to map all clean reads onto the published pineapple genome (Dobin et al., 2013). The resulting BAM files were then processed using StringTie for estimation of expression levels (Pertea et al., 2015), in which the transcripts per kilo base million (TPM) normalized method was employed. The TAU method proposed by Yanai et al. (2005) was applied on the expression matrix to detect genes that were constitutively expressed or expressed in specific tissues (Kryuchkova & Robinson-Rechavi, 2015). Genes with TAU scores lower than 0.2 were defined as constitutively expressed genes while those with TAU scores higher than 0.99 were defined as tissue-specific genes.

WGCNA analysis and KEGG pathway enrichment analysis

To construct a gene co-expression network for pineapple fruit, we employed the WGCNA package (Langfelder & Horvath, 2008). Genes that were not expressed or were expressed at very low levels (TPM lower than 0.1) in all samples were filtered out. The filtered TPM matrix was then imported into WGCNA for analysis. All parameters were kept as default except the following: networkType was set to signed, softPower to 10, minModuleSize to 100, deepSplit to 2.0, and MEDissThres to 0.1. A software package made in-house, TBtools, was used for the Gene Ontology Term and the KEGG pathway enrichment analysis and visualization (Chen et al., 2018).

eFP browser database construction

We used a docker container to construct a pineapple fruit eFP browser that could be used over the long term. The CentOS 7 image was pulled from docker.io, and dependent software packages for the eFP browser were installed according to the manual. The eFP browser installation package was downloaded from http://www.bar.utoronto.ca/efp/development/ and some scripts in the package were adapted for RNAseq data. To facilitate the data importing process, we wrote two python scripts to convert tab-delimited files into SQL command records. After the browser was set up locally, we made a docker image and uploaded it to our VPS (http://www.tbtools.xyz/efp/cgi-bin/efpWeb.cgi) to provide service for all users.

Results

Pineapple fruit detaching and sampling

Pineapple forms a collective fruit that is developed from approximately 100 individual florets. This fruit formation is similar to some berries (Figs. 1A and 1B). Pineapple fruit development accompanies its flower development. In this study we focused on fruit development and defined the time when the first round/batch of flowers started to open as the first fruit developmental stage. The stage when flowering ceases was defined as the second stage. Finally, the third stage was defined as when florets reached mature status and stopped expanding (Fig. 1A).

Figure 1 Three main stages of pineapple fruit development.

(A) Three developmental stages of pineapple fruit. Stage 1 corresponds to the time when flower buds start to bloom and fruit begins to expand in size. At the same time, cell division dominates biological processes in fruit. Stage 2 corresponds to the time when flowering period ends and fruit expands its size at the fastest pace. At the same time, cell expansion dominates biological processes in fruit. Stage 3 corresponds to the time when fruit begins its ripening process and fruit expansion process stopped. (B) Different parts of pineapple fruitlet. Stage 1, Stage 2 and, Stage 3 denote the developmental stage of each fruitlet. Each fruitlet was detaching into bract, flower-disk, sepals, petals, and others. (Photo credit: Qi Mao.)

Pineapple fruits contain several main tissue types: bracts, sepals, flower disks, receptacles, ovaries walls, placentas, ovules, and cores (Fig. 1B). In addition, we also examined flower tissues such as petals, styles, and stamens, as well as vegetative tissues such as leaves, roots, and stems. The berry floret is enlarged as it expands. Petals are colored pale purple. Importantly, the rate of enlargement of the bract and receptacle are faster than the rates of the flower disk and placenta, which results in the formation of pineapple fruit eyes.

The global landscape of the pineapple fruit developmental transcriptome

To characterize expression dynamics throughout pineapple fruit development, RNAseq libraries were constructed and sequenced for 14 different tissues collected at three major developmental stages (Table S1; Fig. S1). Using the Illumina sequencing platform, ∼1.19 billion 125 bp paired-end reads (149 GB) were generated, corresponding to an average of 33 million reads per tissue sample (Table S1). Most reads were mapped onto exon regions, indicating that the RNAseq data was clean and could be used in further analyses (Fig. S1).

The pineapple genome was expected to contain 27,024 genes (Ming et al., 2015). We found that 23,789 (88.02% of the total) annotated genes were expressed in at least one of the 14 tissue types (Figs. 2A and 2B). The detection of expression of such a high proportion of genes may be attributed to the diversity of tissues included in the study as well as high sequencing depth and coverage. The expression of 3,235 gene models (11.98% of the total) were either not detected in any of the tissues or were detected below background levels (TPM lower than 0.1). These gene models may represent pseudogenes or they may be expressed only under specific biotic or abiotic stress conditions. Expression pattern correlation analysis showed that expression profiles of vegetative tissues were more distinct than profiles of reproductive tissues (highlighted in purple and red rectangles in Fig. 2C).

Figure 2 Global gene expression patterns in different tissues of pineapple.

(A) Number and percentage of genes expressed in each of the 14 tissue samples during pineapple fruit development. (B) Heatmap illustration of spearman correlation coefficient between two of each sample. The darker the color, the higher the similarity of gene expression patterns between two samples.

Tissue-specific and constitutive expression dynamics

Genes can be expressed constitutively or preferentially according to their different roles in plant development. To study the mechanisms underlying the development of pineapple collective fruit, we utilized TAU methods to detect housekeeping (TAU scores lower than 0.2) and tissue-specific (TAU scores higher than 0.99) genes. In total, 273 genes were found to be expressed constitutively with varying expression levels (Fig. 3A; Tables S2–S3). In contrast, we found 1,051 tissue-specific genes (Fig. 3B; Tables S4 and S5). Of these, most were expressed preferentially in stamen (286 genes), root (170 genes), and leaf (147 genes) (Fig. 3B; Table 1). Among these genes, most were also stage-specific, and were preferentially expressed in the same tissue at a given development stage (Fig. S2).

Figure 3 Heatmap of consititutive and tissue-specific genes.

(A) Heatmap illustrating the expression pattern of consititutively-expressed genes across 14 tissues. TPM values are log2-tranformed for better visualization. (B) Heatmap showing the expression pattern of specifically expressed genes across 14 tissues. TPM values are normalized by row for better visualization.

Table 1 Number specifically expressing genes of each tissue.

Tissue	Gene Num.	
Bract	77	
Core	29	
Flower_disk	31	
Leaf	147	
Ovary_wall	34	
Ovule	97	
Petal	11	
Placenta	47	
Receptacle	30	
Root	170	
Sepal	42	
Stamen	286	
Stem	33	
Style	17	

Co-expression network analysis

Using the sequencing data generated earlier, we used WGCNA to detect gene co-expression during pineapple fruit development. In total, 18 gene modules were classified, with the number of genes involved ranging from 155 (light green module) to 4,522 (turquoise module) (Fig. 4A; Table S6). Among these modules, several showed spatial and temporal expression patterns (Fig. S3). Genes of the red module were preferentially expressed in bracts and sepals. KEGG pathway enrichment analysis indicated that the functions of the genes of the red module were associated with plant defense, including plant-pathogen interactions and MAPK signaling (Figs. 4B; Fig. S4). In reproductive tissues, such as receptacles, stamens, and styles, genes of the salmon-colored module were highly expressed at the beginning of fruit expansion and faded afterward. These genes were generally not expressed in vegetative tissues such as leaves and roots. According to pathway enrichment analysis, these genes were involved in endocytosis, transport and catabolism, and cutin suberine and wax biosynthesis (Figs. 4C; Fig. S5). Genes belonging to the light cyan module were also expressed at lower levels in vegetative tissues, however, in reproductive tissues, they showed expression patterns similar to those of the genes in the salmon-colored group. As indicated by gene set enrichment analysis, genes involved in terpenoids and polyketides metabolism were over-represented in this group (Fig. 4D; Fig. S6).

Figure 4 Dendrogram of co-expressing gene modules and KEGG pathways enrichment bar plot of three modules with interesting expression pattern.

(A) In the upper panel, a dendrogram showed all genes imported for WGCNA analysis. Each vertical line corresponds to one gene. In the bottom panel, rectangles of different color denoted the classification of all genes. (B–D) KEGG pathway enrichment bar plot of red, salmon and lightcyan module. P-value were log10-transformed for better visualization.

The Pineapple eFP browser for fruit development analysis

Electronic fluorescent pictograph browsers can accelerate routine gene expression analysis, especially in developmental biology. To facilitate visual exploration of gene expression during pineapple fruit development, we leveraged docker technology to build a pineapple eFP browser (http://www.tbtools.xyz/efp/cgi-bin/efpWeb.cgi). The browser provides a user-friendly web-based interface. Users can enter a specific pineapple gene id and obtain information on its expression in a short time. To ensure that the pineapple eFP browser was working correctly, we manually checked the gene expression pattern graphs of three different genes. Three genes (Aco031614, Aco023419, and Aco004566) were used for this test. Figure 5B uses darker colors to denote higher expression levels of the query gene in specific tissues. This figure shows that a randomly picked gene (Aco031614) is expressed at the highest levels in bracts, followed by sepals, petals, core, and others (Fig. 5B). A stamen-specific gene (Aco023419) was shown in an image in which only stamens were colored red; all other tissues were colored bright yellow (Fig. 5C). In contrast, all tissues were colored red when testing a constitutively expressed gene (Aco004566) (Fig. 5D).

Figure 5 Pineapple eFP browser views of different gene query.

(A) Default view of pineapple eFP browser with different color denoting different tissues. (B) A view of a randomly picked gene (Aco031614) with darken color denoting higher expression levels. (C) A view of a stamen-specific gene (Aco023419). (D) A view of a constitutively expressed gene (Aco004566).

Discussion

We obtained a comprehensive and high-quality fruit transcriptome for pineapple fruit

Although extensive RNA sequencing data for pineapple exists on public repositories such as NCBI SRA, only a few of these were produced from fruit (Ong, Voo & Kumar, 2012; Ma et al., 2015; Ming et al., 2015; Chen et al., 2016; Liu & Fan, 2016; Redwan, Saidin & Vijay Kumar, 2016). No data is available for the different tissues present in the pineapple collective fruit, and this fact hinders studies of pineapple fruit development. Here, we present 36 libraries of pineapple, covering all distinct tissues of its fruit at three developmental stages. Almost 90% of the genes were detected in the current analysis based on gene strucutre annotation from the reference pineapple genome. The difference in the number of genes detected in each tissue indicated that many genes are expressed preferentially in some tissues and not at all in others. In order to generate a transcriptional landscape of pineapple for further study, no replicate has been set. To ensure the data is trusted, we have applied qRT-PCR to validate expression of two pineapple gene families in our previous work (Liu et al., 2017; Xie et al., 2018).

Detection of constitutively expressed and tissue-specific genes provide the basis for further study of pineapple fruit development

Tissue-specific genes play fundamental roles in tissue differentiation and maintenance (Pattison et al., 2015). Detection of tissue-specific genes in pineapple fruit tissues can provide the basis for further study of genetic mechanisms responsible for fruit phenotypes, and may also provide a basis for genetic manipulation. For instance, in collaboration with a developed pineapple genetic-transformation system (Soneji & Nageswara, 2009), we can express a gene of interest in a specific tissue with the promoter sequence of a tissue-specific gene. In this study, we identified ∼1,000 tissue-specific genes. Among these, stamens possessed the most tissue-specific genes; this may be expected since many pollen-specific genes have already been characterized. In addition, leaf and root tissues both possessed more than 100 tissue-specific genes; this is almost twice the number of genes in fruit tissues. This may be due to that these genes are involved in nutrient uptake.

In contrast, only a few genes were preferentially expressed in tissues of the pineapple fruit. a total of 29 genes were core-specific genes. Among them, Aco002250 coding a copper ion binding protein possessed the highest expression level. Its homology, LOC_Os01g32790, in rice is preferentially expressed in inflorescence according to the records in the MSU Rice database (Kawahara et al., 2013). A Chalcone synthase (CHS) gene, Aco008872, was also found to be core-specific. In maize, silencing CHS increases Lignin Content (Eloy et al., 2017). The pineapple fruit core is the extension of the stem, which is rich in lignin. As the fruit develops, the CHS gene specifically expressed in the core might take part in reducing the lignin content, making the core more edible. In total, we detected 31 genes preferentially expressed in flower-disk. Most of them were annotated as “hypothetical protein.” One auxin response factor might be useful for further study, whose homology, AT1G30330, in Arabidopsis is preferentially expressed carpel according to eFP browser on TAIR (www.arabidopsis.org). Pineapple flower-disk decides the depth of the fruit-eye. Controlling expression pattern of pineapple ARF6 might be a possible way to breeding pineapples for shallow fruit-eyes. A total of 34 genes were found to be ovary-wall specific genes. It is interesting that, out of them, two shikimate kinase gene, Aco004125 and Aco029719, possessed the highest expression levels. Shikimate kinase is the key component of the shikimate pathway, from where a vast number of aromatic compounds originated (Bentley & Haslam, 1990). Pineapple is famous for its special fragrance. The high expression level and the preferential expression pattern of these two shikimate kinase genes in ovary-wall indicate that pineapple fragrance may be generated from ovary-wall and these two genes are involved in fragrance production of pineapple. In total, 47 genes are preferentially expressed in pineapple placenta. Among them, we found Aco006346 coding a homology of sugar transporter SWEET17. In Arabidopsis, SWEET17 mediates fructose transport across the tonoplast (Guo et al., 2014). Pineapple SWEET17 might also play a similar role during fruit development. Apart from this, we also identified 273 constitutively expressed genes, and these may serve as useful reference genes for qRT-PCR experiments of pineapple fruit.

Gene module classification can facilitate further study of pineapple fruit

Genes function as networks in all biological processes. To uncover the mechanisms underlying different traits, network-level analyses are required. Here, we classified all expressed pineapple genes into 18 gene modules. Among these, genes of the red module were preferentially expressed in bracts and sepals. According to the KEGG pathway enrichment result, most members were associated with plant defense, including plant-pathogen interactions and MAPK signaling. As the pineapple fruit develops, it faces various kinds of bio-stress like pest binding. Thus, it is possible that these resistance-associated genes are expressed with higher levels in the outer part of pineapple fruit to defense the adversity. Genes of salmon module might have similar functions but function in a different way. These genes, functionally related to cutin, suberine, and wax biosynthesis, are only highly expressed at the beginning of fruit expansion. They may be important for all reproductive tissues of pineapple but only functions at the specific stage. In contrast, genes in the light cyan module seems to be linked to fruit quality. Most members of this group are involved in terpenoid and polyketide metabolism. It is likely that this module contains key genes related to pineapple ripening and fragrance biogenesis. In addition, other modules also seem to have distinct expression profiles and apparent functions. Together, these modules can be used for gene screening and further experimentation.

The eFP browser is an important tool for pineapple fruit study

Studies of fruit are important for fruit tree breeding and research, since different crops form different fruit types. Tomato, the most famous fruit study model species (Kimura & Sinha, 2008), bears simple fruit developed from ovaries. Strawberry, another model fruit species (Li et al., 2011), possesses aggregate accessory fruit developed from receptacles (Hollender et al., 2012). Both species have eFP browsers, which provide researchers with a simple method to explore gene expression data. In the present study, we built a pineapple eFP browser in which distinct pineapple tissues are colour-coded. Compared to traditional gene expression databases, which visualized gene expression matrices as tile plots (Nussbaumer et al., 2014), users can explore specific genes in a cartoon image and see how expression levels differ among pineapple tissues. Moreover, the browser leverages docker technology; with docker images, users can also re-build their own pineapple fruit eFP browsers locally relatively easily in multiple operating systems. In this way, the pineapple eFP browser can be accessible without a network connection.

Conclusions

In the present study, we provide a comprehensive and high quality pineapple fruit transcriptome. We have characterized thousands of tissue-specific genes as well as 18 gene co-expression modules. Moreover, a pineapple fruit eFP browser has been constructed to explore gene expression in pineapple fruit and to facilitate further study of pineapple fruit development.

Supplemental Information

Supplemental Information 1 Summary statistics for transcriptome sequencing of various tissues in genome-wide developmental expression analysis of pineapple.

Click here for additional data file.

Supplemental Information 2 Expression patterns of constitutively expressing genes of pineapple.

Click here for additional data file.

Supplemental Information 3 Expression patterns of constitutively expressing genes of pineapple during fruit development.

Click here for additional data file.

Supplemental Information 4 Expression patterns of specific-expressing genes of pineapple.

Click here for additional data file.

Supplemental Information 5 Expression patterns of specific-expressing genes of pineapple during fruit development.

Click here for additional data file.

Supplemental Information 6 Co-expressing analysis result of the pineapple transcriptome during fruit development.

Click here for additional data file.

Supplemental Information 7 Summary statistics for transcriptome sequencing of various tissues in genome-wide developmental expression analysis of pineapple.

Click here for additional data file.

Supplemental Information 8 Expression patterns of constitutively expressing genes of pineapple.

Click here for additional data file.

Supplemental Information 9 Expression patterns of constitutively expressing genes of pineapple during fruit development.

Click here for additional data file.

Supplemental Information 10 Expression patterns of specific-expressing genes of pineapple fragment.

Click here for additional data file.

Supplemental Information 11 Expression patterns of specific-expressing genes of pineapple during fruit development.

Click here for additional data file.

Supplemental Information 12 Co-expressing analysis result of the pineapple transcriptome during fruit development.

Click here for additional data file.

We thank all members of the He lab for their help in separating different pineapple fruit tissues.

Additional Information and Declarations

Competing Interests

Author Contributions

Data Availability

The authors declare that they have no competing interests.

Qi Mao conceived and designed the experiments, performed the experiments, prepared figures and/or tables, authored or reviewed drafts of the paper, approved the final draft.

Chengjie Chen conceived and designed the experiments, analyzed the data, prepared figures and/or tables, authored or reviewed drafts of the paper, approved the final draft.

Tao Xie performed the experiments, contributed reagents/materials/analysis tools, approved the final draft.

Aiping Luan contributed reagents/materials/analysis tools, approved the final draft.

Chaoyang Liu contributed reagents/materials/analysis tools, approved the final draft.

Yehua He conceived and designed the experiments, approved the final draft.

The following information was supplied regarding data availability:

All raw data used for the transcriptome analysis and database construction are available at Sequence Read Archive (SRA) at NCBI under Project ID PRJNA483249.

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
