# Peer review of "Comprehensive tissue-specific transcriptome profiling of pineapple (Ananas comosus) and building an eFP-browser for further study"

_PeerJ, doi:10.7717/peerj.6028_

## Round 0.1 · original submission · Major Revisions

Your manuscript has been seen by two qualified reviewers. Based on their detailed assessment and my own, I feel this manuscript is suitable for publication in PeerJ following a series of major revisions. The lack of biological replicates is a major limitation and should be addressed in a revision. Ideally, more biological replicates should be collected for each developmental stage, but this may be beyond the scope of this manuscript.

Reviewer 1 ·

Basic reporting

See below #4

Experimental design

Major problem is lack of biological replications for each stage and tissue.

Validity of the findings

replication is an issue though if addition pineapple transcriptome tissue data can be added to the eFB then this manuscript has merit

Additional comments

The difficulty in the results are the absence of biological replication making analysis a challenge.

ABSTRACT:
Pineapple is a tropical plant and can only be grown in the warm subtropics
Line #26 - briefly the stages of fruit growth when fruit were sampled: early flowering, late flowering and mature green.
Figure 1A claims that Stage #3 is when the fruit is starting to ripen however the picture show no yellowing between the eyes at the base of the fruit. The manuscript gives no details as to the variety of this fruit though it is not a Smooth Cayenne Clone possibly “Queen”.


INTRODUCTION
Line #39 - Pineapples do not grown on trees - the plant is a herbaceous perennial. Replace tree with herb.

Line #42 - The whole fruit is not consumed, only the fleshy coalesced floral parts as shown in Figure 5.

MATERIALS and METHODS
The word “fruits” implies different types of fruit (lizhi, apples, oranges) when refering to one type of fruit such as pineapple “fruit” is both singular and plural. Please correct usage in the text.
Line #81 give the name of the variety used in this study.
Line 92-92 not sure of the relevance of this last sentence.
Line 101 - “Tau method” is this the Kendall-tau rank correlation coefficients - clarify or is it Yanai et al, 2005 approach?
Line #109 - First use of TPM (Transcripts per Kilo base Million) as normalization method.

RESULTS
Line 135 - “fruits” should be individual “florets”
Line 167 - TAU methods this ranges from 0 to 1 if I recall broad to specific expression may want to explain to reader.
Line 214 - Pattison et al., 2015 did there research on tomatoes not pineapple - correct.
Line 248- Strawberry is not an aggregate fruit, more correctly it is a aggregate accessory fruit as we eat the swollen flower receptacle,

Reviewer 2 ·

Basic reporting

The paper titled “Comprehensive tissue-specific transcriptome profiling of pineapple (Ananas comosus) and building an eFP-browser for further study ” reports a transcriptomic resource that has the potential to be beneficial to the subtropical fruit research community. The authors generated RNA-sequencing data at three stages and 14 different tissues of early fruit (and development (and flower parts) in order to identify stage and tissue-specific genes. The authors use the Tau method to identify tissue specific genes. Additionally, WGCNA was used to establish clusters of co-expressed genes. This work is important for researchers interested in fruit development and the inclusion of the eFP makes these data much more attainable. However, the transcriptomic work is limited by single biological sampling at each stage-tissue.

(1) Issues with biological replicates
The authors spend a large part of their paper discussing the different stages of development sampled. It is clear from Figure 1 that stage 1, stage 2 and stage 3 are drastically different from each other in terms of development. Unfortunately, no differential expression was conducted across these three stages possibly due to only one biological replicate at each tissue and time point.

Instead the authors treat these three stages of the same tissue as three replicates in order to identify tissue-specific expression. This treatment may result in unsuitable representation for genes only expressed at early or late stages of fruit development. For example, gene Aco001279 in the browser has a standard deviation of 62.79 in the floral disk; this high standard deviation indicates the data points are spread out over a very wide range of values. Collapsing such diverse datasets makes these data difficult to interpret and valuable stage-specific information is lost.

(2) The authors concluded that there were 1,051 tissue specific genes but did no perform other analysis on categorizing these genes or their potential role in fruit development. Was GO enrichment done on these tissue specific genes? Do the function of these genes corroborate with their tissue specificity? The authors also concluded that they classified 18 gene modules. They did GO annotation on these genes but they didn't seem to have looked any further.

(3) The legends in many of the figures are unclear. Specifically, I found the legend for 2A difficult to understand and a redundant representation of Figure 2B. The distance metric used for Figure 2C should be included. In Figure 3, there is no mention of what the values in the heatmap actually represent (tpm, z-score, rpkm?). There is also no labeling on the scales on within the supplemental figures. The color scheme for the supplemental figure heatmaps is red and green which would be unacceptable for readers with red-green color blindness.

(4) Some small notes:
On line 70, it should be “eleven” not “eight” pineapple fruit samples
On line 65 you only cite the strawberry eFP but also mention the tomato and grape eFP
On line 98 Trimmomatics should be Trimmomatic
On line 111 defaults should be default

In conclusion, a lack of stage-specific differential expression limits the utility and value of the work, including eFP. The lack of replicates makes this data statistically unsound to accomplish these necessary components. Additionally, a more in-depth analysis is needed for the Tau and WGCNA segments.

Experimental design

no comment

Validity of the findings

no comment

Additional comments

no comment

---

## Round 0.2 · Minor Revisions

Based on the reviewers detailed assessment and my own, I feel this manuscript is well suited for publication in PeerJ pending a few remaining minor revisions.

Reviewer 1 ·

Basic reporting

The authors have improved the presentation and removed the ambiguities in the first version. The major concern is still the absence of biological replication, though quantitative PCR does give support to their conclusions.

The pineapple eFP browser is an excellent development from the non-replicated data presented and provides a structure to add results from future transcriptome profiling. The author may want to consider a statement how additional data can be added to build upon this effort..

Experimental design

Biological replication is still the major problem

Validity of the findings

Biological replication is a problem

Additional comments

None

Reviewer 2 ·

Basic reporting

The authors have addressed most of previous criticisms except the issue with replicates. A lack of funding or limited funding is not the reason for sacrificing the quality of the research. Given that eFP will be a useful resource for the community, I am OK to let this pass as long as the authors provide warnings (in their website and paper) to the eFP users concerning the lack of replicates and suggesting the users to validate the data using qRT-PCR.

There is a recent publication in Horticulture Research on Pineapple Genomics Database (PGD) by Xu et al., 2018. The authors of this manuscript should cite the PGD paper and discuss what sets this manuscript apart from PGD.

There are still issues that should be address:
(1) Is pineapple an accessary fruit or botanical fruit or both? The statement “All fleshy coalesced floral parts of the pineapple inflorescence” does not make any sense to me. Does the fleshy fruit consist of an inflorescence plus all flowers? Which part of the flower? Ovary wall (the botanical fruit) or the entire flower?
(2) Please label each dissected floral part in Figure 1B.
(3) Please also label Figure 1A, indicating which is bract, which is sepal, and which is leaf.
(4) Is stage 1 pre-fertilization or post-fertilization? If pre-fertilization, fruit should not expand at stage 1 unless pineapple is different. Please make this clear in text and in figure legend.
(5) There are grammar mistakes in many areas, especially in figure legend and in the newly added Discussion. For instance, in Figure 2B legend: “Darker the color denotes, higher similarity the global gene expressing patterns between two samples”. You mean “The darker the color, the higher the similarity between two samples”?
(6) In many places (legends and Discussion), you use “genes preferentially expressing….”, “genes specifically expressing…”, It should be “genes preferentially expressed….”.
(7) In Figure 3 legend, “constitutively-expressing genes…” should be “constitutively expressed genes…”.
(8) In Figure 3 legend, “log-transformed”, you mean log10 or log2 transformed?
(9) Figure 3 legend, “normalized-scaled by row..”, you mean “normalized by row”?
(10) In Figure 4 legend, “interesting expressing pattern” should be “interesting expression pattern”.
(11) In Figure 5 (eFP), the scale label is too small to see.

Experimental design

no comment

Validity of the findings

no comment

---

## Round 0.3 · accepted · Accept

Your revised manuscript has addressed the previous concerns expressed by the reviewers and is well suited for publication in PeerJ.

#